# Efficacy and Safety of Dose-Dense Chemotherapy in Breast Cancer: Real Clinical Data and Literature Review

**DOI:** 10.3390/curroncol32080441

**Published:** 2025-08-06

**Authors:** Keiko Yanagihara, Masato Yoshida, Tamami Yamakawa, Sena Kato, Miki Tamura, Koji Nagata

**Affiliations:** 1Department of Breast Surgery and Oncology, Nippon Medical School Tama-Nagayama Hospital, 1-7-1 Nagayama, Tama-shi, Tokyo 206-8512, Japan; s13-102yt@nms.ac.jp (T.Y.); s14-031ks@nms.ac.jp (S.K.); t-miki@nms.ac.jp (M.T.); 2Department of Pharmacy, Nippon Medical School Tama-Nagayama Hospital, 1-7-1 Nagayama, Tama-shi, Tokyo 206-8512, Japan; yoshida-m@nms.ac.jp; 3Department of Pathology, Nippon Medical School Tama-Nagayama Hospital, 1-7-1 Nagayama, Tama-shi, Tokyo 206-8512, Japan; k-nagata@nms.ac.jp

**Keywords:** dose-dense chemotherapy, early breast cancer, neoadjuvant chemotherapy

## Abstract

Breast cancer treatment varies based on tumor characteristics and recurrence risk. Dose-dense chemotherapy is a strategy that shortens the interval between chemotherapy cycles to improve outcomes. This study evaluated real-world data from 80 breast cancer patients treated with this method. The results showed that patients with triple-negative breast cancer responded particularly well, with a high rate of complete tumor disappearance. Common adverse events included fatigue, reduced blood counts, and nerve-related symptoms, which were mostly manageable. However, a few patients experienced more serious complications such as pneumonia or low platelet levels. Over 80% of patients completed the treatment as planned. Based on the results of this study, dose-dense chemotherapy may offer significant benefits for patients with high-risk or aggressive breast cancer. These findings support its use in carefully selected cases and may contribute to more personalized and effective treatment strategies in early breast cancer care.

## 1. Introduction

Breast cancer has been shown to have distinct characteristics in terms of recurrence and prognosis depending on its subtype. Morgan E et al. reported in a review of studies involving 280,000 breast cancer patients that regardless of the follow-up period, hormone-receptor-negative patients had a higher risk of recurrence than hormone-receptor-positive patients [1]. In particular, triple-negative breast cancer (TNBC), which is estrogen receptor (ER)-negative, progesterone receptor (PgR)-negative and human epidermal growth factor receptor 2 (HER2)-negative, has been reported to have higher rates of recurrence and mortality compared to those in other types [2,3]. There is a peak in distant metastasis within three years of diagnosis, and the mortality rate within five years is high [3]. Additionally, in TNBC and HER2-positive breast cancer, where a pathological complete response (pCR) was achieved with neoadjuvant chemotherapy, it has been reported that the disease-free survival (DFS) and overall survival (OS) rates were significantly improved compared to those in the group that did not achieve a pCR [2,4].

Therefore, in neoadjuvant chemotherapy (NAC), a regimen with high pCR rates is being considered. Dose-dense chemotherapy is a treatment strategy that reduces the interval between doses of chemotherapy drugs to inhibit tumor cell repopulation and improve therapeutic efficacy. In breast cancer, dose-dense chemotherapy given every 2 weeks has been reported to increase the pCR rate after neoadjuvant chemotherapy compared to that under standard chemotherapy given every 3 weeks [5]. Furthermore, this method has also been shown to improve the DFS and OS. The Cancer and Leukemia Group B 9741 (CALGB9741) trial showed that the 3-year DFS was 82% in a dose-dense chemotherapy group and 75% in the standard therapy group, and the 3-year OS was 92% vs. 90%, where both were statistically superior in the dose-dense chemotherapy group. In particular, this effect was more pronounced in premenopausal patients [6]. In the PANTHER trial, the 10-year breast-cancer-recurrence-free survival was improved with dose-dense chemotherapy compared to that under the 3-week regimen. A significant benefit was seen in luminal (HR: 0.83) and HER2-positive (HR: 0.53) subgroups but not in triple-negative breast cancer (HR: 1.02) [7].

Filho OM et al. reported that dose-dense chemotherapy improved the DFS in any subtype by 23% (hazard ratio (HR): 0.77) and the OS by 20% (HR: 0.80) compared to that under standard chemotherapy given every 3 weeks; the benefits of dose-dense therapy were seen for ER+ and ER-negative subsets, without a significant interaction between the treatment arm and ER status [8].

Although dose-dense chemotherapy increases adverse events, such as neutropenia, anemia, and fatigue, compared to standard chemotherapy, neutropenia can be prevented using granulocyte-colony stimulating factor (G-CSF) every 2 weeks in dose-dense chemotherapy, and other adverse events are within the acceptable range [5,6,7,8,9,10,11,12,13,14,15,16,17]. Pegfilgrastim was needed as the G-CSF.

In this study, we evaluated the efficacy and safety of dose-dense chemotherapy based on actual clinical data on dose-dense chemotherapy administered at our hospital and compared them with the existing literature.

## 2. Materials and Methods

Eighty breast cancer patients who received dose-dense chemotherapy at our institution over a 5-year period from January 2020 to December 2024 were included in this study. The treatment regimen consisted of four doses of epirubicin and cyclophosphamide (dose-dense EC) followed by four doses of paclitaxel (dose-dense paclitaxel) every two weeks, with pegfilgrastim as the G-CSF. For alcohol-intolerant patients, paclitaxel was replaced with docetaxel every 3 weeks as the taxane drug.

The single dose of dose-dense EC therapy was 90 mg/m^2^ of epirubicin and 600 mg/m^2^ of cyclophosphamide. Dose-dense paclitaxel was administered at a single dose of 175 mg/m^2^. Pegfilgrastim was used in combination with either a body pod or subcutaneous injection. A 3.6 mg body pod was applied on the day of treatment, and the drug was automatically injected subcutaneously 27 h later. A 3.6 mg subcutaneous injection was administered on the second or third day of treatment, with the drug administered subcutaneously.

The efficacy and safety of dose-dense chemotherapy were evaluated retrospectively, and pathologic effects were also analyzed in addition in the NAC group. A pathological complete response was defined as the disappearance of the invasive carcinoma, even if an intraductal component remained. CTCAE 5.0 was used to evaluate adverse events, and the Kaplan–Meier survival curve and Fisher’s exact probability test were used for the statistical analysis. For cases in which docetaxel was used from the outset due to alcohol intolerance, only dose-dense EC therapy was evaluated in this study.

## 3. Results

The patients’ background is shown in Table 1. The age of the patients was 30~78 years (mean: 56.7 years), and the observation period was 3~64 months (median: 38 months). The subtypes were a triple-negative type (TNBC) in 30 cases and a luminal type (luminal) in 50 cases; HER2-positive cases were not included. Fifty-three patients (66.3%) were axillary-lymph-node-positive cases at the time of diagnosis, with clinical stage I in 15 cases, stage II in 46 cases, and stage III in 19 cases. A total of 75 cases were invasive ductal carcinoma of the breast, 4 cases were invasive lobular carcinoma, and 1 case was invasive micropapillary carcinoma.

There were 27 premenopausal patients, and there were 53 postmenopausal patients.

When stratified by menopausal status, the postmenopausal group showed a significantly higher rate of pathologically node-negative disease after surgery compared with that in the premenopausal group (62.3% vs. 25.9%; Fisher’s exact *p* = 0.004; OR for ≥1 positive node in pre- vs. postmenopausal patients: 4.71; 95% CI: 1.69–13.13). Trends toward a lower hormone-receptor-positive rate, a higher Ki-67 index, and a higher proportion of triple-negative tumors were observed in the postmenopausal patients; however, these did not reach statistical significance (all *p* > 0.05). The pathologic complete response rates among patients receiving neoadjuvant chemotherapy were similar between groups (26.3% vs. 30.6%; *p* = 1.00).

Of the 55 patients who received NAC, 22 had TNBC and 33 had luminal-type cancer (Table 2). The overall pCR rates were 29.1% (16/55) and 59.1% (13/22) in the TNBC group and 9.1% (3/33) in the luminal group. The results of the Fisher’s test showed that the pCR rate in the TNBC group was statistically significantly higher than that in the luminal group (odds ratio: 14.4, *p* = 0.0001).

The eight cases of TNBC who did not undergo NAC were those who were considered to have non-invasive carcinoma preoperatively but were found to have invasion on postoperative pathology or those who preferred to proceed with surgery because of the COVID-19 situation. Around 2020, there were cases where surgeries were postponed due to COVID-19 infections among medical staff, patients, and their families. As a result, some patients requested to undergo surgery as early as possible. They were concerned that their surgeries might be canceled due to COVID-19.

We investigated the differences in the pCR rates after NAC between TNBC and the luminal type in premenopausal and postmenopausal patients (Figure 1).

However, no significant differences in treatment efficacy were observed between the premenopausal and postmenopausal patients across subtypes. When comparing TNBC before and after menopause using Fisher’s exact probability test (two groups × 2), the following results were yielded: *p* = 1.00 (two-tailed), odds ratio (pre/post) = 1.56 (95% CI: 0.22–11.09), and risk ratio (RR) = 1.19 (95% CI: 0.58–2.42). When limited to luminal-type cancer, *p* = 0.55, OR = 3.45, and RR= 3.08 (95% CI: 0.31–30.59). For pre- and postmenopausal women with a mixed subtype, *p* = 0.77, and OR = 1.20 (95% CI: 0.36–4.03). In either case, no results were obtained indicating that premenopause was statistically advantageous over postmenopause. As shown in Table 2, the only finding was that TNBC had a better treatment response than that for the luminal type, regardless of menopausal status.

In either case, no results were obtained indicating that premenopause was statistically advantageous over postmenopause.

The Kaplan–Meier curves for the DFS are shown (Figure 2). The number of recurrence events was only five cases in the postmenopausal group, and there was no recurrence in the premenopausal group, and no significant difference was observed. There were only three deaths after menopause, and no trend was observed before or after menopause.

The treatment completion rate was 82.5% (66/80), and 14 patients (17.5%) could not continue dose-dense chemotherapy due to adverse events. The main adverse events are shown in Table 3.

Hematologic adverse events, such as liver dysfunction (58.8%) and anemia (57.6%), were observed in more than half of the patients, but most were Grade 1 or 2. Thrombocytopenia was observed in 22.6% of patients, and leukopenia and neutropenia were observed in 18.8% of patients. Febrile neutropenia was observed in seven patients (8.8%).

Non-hematologic complications included alopecia (100%), myalgia (70.1%), arthralgia (57.6%), and peripheral neuropathy (55.0%), which were observed in more than half of the patients. In addition, fatigue (43.8%), nausea (37.6%), and anorexia (27.6%) were also common.

Grade 3 or higher adverse events included leukopenia and neutropenia at 11.3%, febrile neutropenia at 8.8%, and thrombocytopenia at 1.3%. Non-hematologic adverse events included pneumonia at 5.0% and fatigue at 2.5%, while anorexia, nausea, arthralgia, and myalgia were each experienced in 1.3%. Among these, Grade 4 events consisted of leukopenia (5.0%), neutropenia (5.0%), and thrombocytopenia (5.0%). The reasons for discontinuation were Grade 3/4 anemia and neutropenia, drug-induced pneumonia, pneumocystis pneumonia, liver dysfunction, and Grade 4 thrombocytopenia due to pegfilgrastim.

Two of the patients who discontinued treatment were diagnosed with pneumocystis pneumonia, which needed to be distinguished from COVID-19-associated pneumonia. Due to severe pneumonia, oxygen administration and hospitalization for more than two weeks were required. After discharge, continued chemotherapy became difficult due to a decline in performance status, so chemotherapy was not continued, and surgery was performed.

One patient diagnosed with thrombocytopenia due to pegfilgrastim was treated with a pegfilgrastim body pod in the first instance, and Grade 4 neutropenia and thrombocytopenia developed on day 8. Antibiotic therapy was administered for febrile neutropenia, and their platelet counts decreased to 23,000/mm^3^, but no blood transfusion was given because the patient refused. The patient was kept under observation, and improvement to Grade 1 was observed by day 15. We also switched to subcutaneous injection of pegfilgrastim for this patient and reduced the dose of chemotherapy. Neutropenia improved, but thrombocytopenia similarly decreased to Grade 4. Considering thrombocytopenia associated with pegfilgrastim, we switched to a 3-week regimen without pegfilgrastim. This means that dose-dense chemotherapy was discontinued. After this change, thrombocytopenia remained at Grade 1. In our institution, several thin women with low subcutaneous fat who received a pegfilgrastim body pod developed Grade 4 febrile neutropenia. We switched to subcutaneous injections of pegfilgrastim, and Grade 4 neutropenia no longer occurred.

Other adverse events, such as liver dysfunction, myalgia/arthralgia, peripheral neuropathy, fatigue/malaise, and alopecia, occurred, with most cases showing symptomatic improvements after chemotherapy was completed.

There were three cases of distant recurrence (3.8%), including one case of inflammatory breast carcinoma with TNBC in whom a pCR was obtained through NAC. Two months after surgery, multiple liver metastases were found, and the breast carcinoma was PD-L1-positive. Chemotherapy with an immune checkpoint inhibitor was ineffective, and the patient died four months after surgery. The other two patients had luminal-type carcinoma with lymph node metastasis. Both patients had a non-pCR after neoadjuvant chemotherapy and died two years later due to liver and lung metastases, respectively.

## 4. Discussion

Dose-dense chemotherapy, characterized by standard-dose agents administered at shortened intervals with G-CSF support, has been shown to improve the disease control in patients with high-risk early breast cancer [2,4,6,8]. The CALGB9741 trial demonstrated significant survival benefits, including a 7% absolute improvement in the 3-year DFS and a 2% improvement in the OS, with pronounced efficacy in premenopausal women [6]. Several subsequent randomized trials and meta-analyses, including those by Venturini, Del Mastro, and Zhou, have confirmed these findings, establishing dose-dense chemotherapy as a standard strategy in selected populations [5,6,7,8,9,10,11,12,13,14,15,16,17]. In particular, Fornier M et al. reported in their review that dose-dense chemotherapy is expected to be particularly effective in triple-negative breast cancer and high-risk premenopausal patients [11]. Furthermore, dose-dense chemotherapy has been reported to improve the DFS and OS compared to those under chemotherapy every three weeks, even in ER-positive triple-negative breast cancer [8].

In our study, too, the TNBC group was more likely to obtain a pCR, with a significant difference of 59% compared to that in the luminal group, which had a pCR rate of 6%. This result supports the high efficacy of dose-dense chemotherapy in TNBC. Although the number of cases was limited, the pCR rate for TNBC patients in this study was 59.1%, which is higher than the pCR rates shown in other clinical trials (32.7–52%) [2,5]. On the other hand, the pCR rate for the luminal type was low at 9.1%, suggesting that the efficacy of dose-dense chemotherapy in luminal-type cancer is limited.

Previous reports have indicated that in TNBC, the presence of insulin-like growth factor II mRNA-binding protein 3 (IMP3) expression is associated with a poor response to chemotherapy [18]. However, in recent regimens, including dose-dense chemotherapy, there has been no difference in the efficacy of preoperative chemotherapy based on the presence or absence of IMP3 expression, and dose-dense chemotherapy is considered effective, even for more aggressive TNBC unresponsive to previous regimens [19].

Adverse events were observed in many cases in the above clinical trials, similar to those seen with conventional chemotherapy. The 2022 Japanese Guidelines for the Management of Breast Cancer provide a literature review of the adverse events associated with dose-dense chemotherapy [20]. Anemia was analyzed in two randomized controlled trials (RCTs) involving 4172 patients. In all grades, the incidence of the development of Grade 1 anemia or higher during treatment was 37.1% in the control group who underwent dose-dense chemotherapy, with approximately 34.5% reaching Grade 3 and 37.9% requiring a blood transfusion. Based on these findings, it is considered necessary to carefully monitor blood sampling data [20,21,22].

In our hospital, anemia during dose-dense chemotherapy improved naturally in all cases upon the completion of chemotherapy. In the above Japanese guidelines for the treatment of breast cancer, the evaluation of febrile neutropenia was analyzed in one RCT involving 2155 participants. The risk was reduced in the dose-dense chemotherapy group (1.4%) compared to that in the control group (3.6%) (risk difference: −0.02; 95% CI: −0.03 to −0.01) [20]. This result was considered necessary due to the mandatory use of pegfilgrastim.

The incidence of adverse events was similar in our study, but febrile neutropenia was observed in 8.8% of cases (7/80). They required antibiotic treatment and hospitalization. At our institution, febrile neutropenia occurred primarily after the first administration, and although the number of cases was small, it was observed in several cases in women who were thin with little subcutaneous fat and who received a pegfilgrastim body pod.

In two cases of pneumocystis pneumonia, febrile neutropenia occurred when the treatment was switched from anthracycline (four instances of dose-dense EC) to taxane (dose-dense paclitaxel). This type of pneumonia is also known as an opportunistic infection, and it was thought that the administration of steroids every two weeks to prevent adverse events may have contributed to its development. Of these, one case was reported by Yagi [23]. Pneumonia caused by Pneumocystis jirovecii pneumonia can appear to worsen despite effective treatment, and this coincided with the spread of COVID-19, making diagnosis and treatment difficult. During the treatment for Pneumocystis jirovecii pneumonia, the phenomenon of symptoms appearing to worsen despite effective treatment is known as a “paradoxical response” or “immune reconstitution inflammatory syndrome (IRIS)” [24].

In this study, Grade 4 thrombocytopenia was observed. A pegfilgrastim body pod was used in the first session of chemotherapy, and the treatment was switched to a subcutaneous injection of a pegfilgrastim formulation for the second session, both of which resulted in Grade 4 thrombocytopenia. After discontinuing pegfilgrastim due to concerns about its effects, thrombocytopenia remained at Grade 1, leading us to conclude that pegfilgrastim was the cause.

Pegfilgrastim stimulates the hematopoietic stem cells in the bone marrow to promote the production of neutrophils. It is modified using PEG (polyethylene glycol), which slows its breakdown in the body and gives it a longer duration of action [25,26,27,28,29,30]. In a dose-dense regimen, it is important to manage neutropenia, and pegfilgrastim is a drug necessary to suppress neutropenia in order for treatment to be administered every two weeks. However, patients using pegfilgrastim have a significantly higher risk of thrombocytopenia compared to that in non-users (adjusted odds ratio: 5.7; 95% confidence interval: 4.3–7.5) [31]. Additionally, an analysis comparing the adverse events between filgrastim and pegfilgrastim indicated that pegfilgrastim was associated with a higher incidence of thrombocytopenia.

Dose-dense chemotherapy has been reported to cause other adverse events, such as anemia, liver dysfunction, fatigue, muscle and joint pain, and peripheral neuropathy [1,2,3,4,5,6,7,8,12]. In this study, similar symptoms were observed, but many cases showed improvements after the completion of dose-dense chemotherapy. Regarding hair loss, our clinic does not use scalp-cooling devices, and this was approved by all patients.

Although the number of cases examined was small, our dose-dense chemotherapy showed a high pCR rate, especially in patients with TNBC. On the other hand, the limited efficacy of dose-dense chemotherapy in luminal-type cancer suggests the importance of an individualized treatment strategy.

Regarding adverse events, the treatment completion rate was 82.5%, indicating that dose-dense chemotherapy was generally well tolerated. However, it is important to note that febrile neutropenia can occur in less than 10% of cases. Some serious complications such as pneumocystis pneumonia and pegfilgrastim-induced thrombocytopenia can rarely occur. By carefully managing adverse events, dose-dense chemotherapy for early breast cancer patients with a high recurrence risk was completed in more than 80% of cases. Particularly in TNBC, it showed a high pCR rate and is considered to have promising therapeutic effects.

This study included several patients aged 70 years or older. Most were able to complete dose-dense therapy similarly to patients in other age groups; however, some elderly patients had concomitant organ dysfunction due to aging.

Yildirim et al. reported that elderly patients aged 65 years or older are more prone to Grade 3–4 adverse events than younger patients (71% vs. 46.4%, *p*  <  0.001), and the pCR rate is lower than that in younger patients (26.6% vs. 33.3%, *p* = 0.24). However, in summary, it is applicable if used selectively with caution regarding its toxicity [32]. Another report states that dose-dense chemotherapy is possible even in elderly patients aged 60 years or older, with toxicity within the acceptable limits [31,32,33].

In high-risk early triple-negative breast cancer, the combination of pembrolizumab, an immune checkpoint inhibitor, and chemotherapy as neoadjuvant therapy has been reported to increase pCR rates (the KEYNOTE-522 study) [34]. Immune checkpoint inhibitors require caution regarding immune-related adverse events, and in facilities like ours with a high proportion of elderly patients, careful consideration is necessary when determining their appropriate use.

CDK4/6 inhibitors plus endocrine therapy is the first-line treatment for hormone receptor-positive, HER2-negative breast cancer with metastatic recurrence. Clinical trials have reported its efficacy as neoadjuvant therapy for early-stage breast cancer, but it is not covered by insurance in Japan, and results from future clinical trials are anticipated.

In this study, it was demonstrated that dose-dense chemotherapy was safe, with acceptable toxicity in a general clinical setting, although caution is required. Additionally, a higher pCR rate was observed in TNBC compared to that in luminal-type cancer. Furthermore, when comparing premenopausal and postmenopausal patients, fewer events were observed in the former, and no significant differences were found in DFS or OS.

## Figures and Tables

**Figure 1 curroncol-32-00441-f001:**
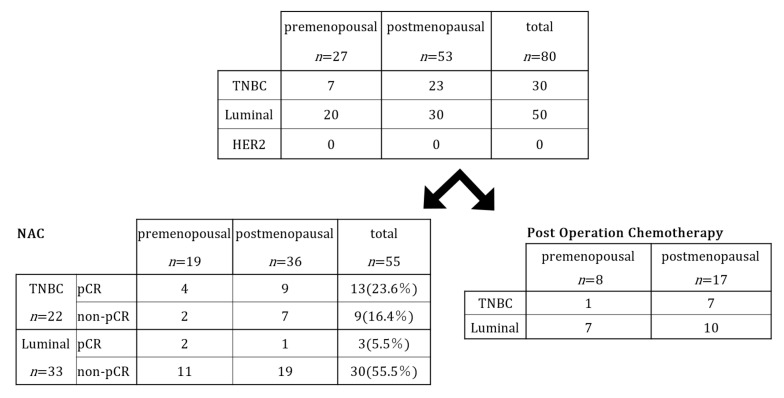
Classification based on menopausal status.

**Figure 2 curroncol-32-00441-f002:**
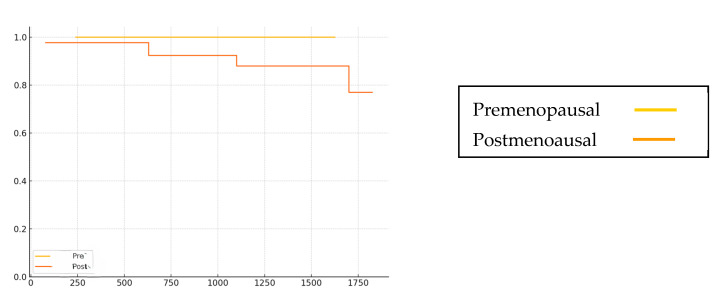
DFS in pre- and postmenopausal patients.

**Table 1 curroncol-32-00441-t001:** Patients’ characteristics.

	Premenopause 27 (33.8%)	Postmenopause 53 (66.3%)	Total 80 (100%)
Median age (range)	44.1 (30–50)	63.0 (46–78)	56.7 (30–78)
Observation period (months)	35 (9–52)	40 (3–61)	38 (3–64)
Menopausal status			
Premenopausal	25	0	25 (31.2%)
Postmenopausal	0	51	51 (63.8%)
Unknown *^1^	2	2	4 (5.0%)
Histopathological type			
Invasive ductal carcinoma	25	50	75 (93.8%)
Invasive lobular carcinoma	2	2	4 (5.0%)
Other	0	1	1 (1.3%)
Clinical stage (before chemotherapy)			
I	4	11	15 (18.8%)
II	16	30	46 (57.5%)
III	7	12	19 (23.8%)
Tumor stage (before chemotherapy)			
T1	6	18	24 (30.0%)
T2	15	21	36 (45.0%)
T3	3	4	7 (8.8%)
T4	3	10	13 (16.3%)
Tumor grade (before chemotherapy)			
1	12	16	28 (35.0%)
2	10	24	34 (42.5%)
3	5	13	18 (22.5%)
Hormone receptor			
ER and/or PgR: positive	20	30	50 (62.5%)
ER and PgR: negative	7	23	30 (37.5%)
HER2			
positive	0	0	0
negative	27	53	80 (100%)
Ki67			
≤20	15	20	35 (43.8%)
>20	12	33	45 (56.3%)
Subtype			
Luminal	20	30	50 (62.5%)
Triple-negative	7	23	30 (37.5%)
HER2	0	0	0
Axillary lymph node status (at the diagnosis) *^2^			
Positive	16	37	53 (66.3%)
Negative	11	16	27 (33.8%)
No. of positive nodes (post-surgery)			
0	7	33	40 (50.0%)
1–3	13	13	26 (32.5%)
4–9	5	6	11 (13.8%)
≥10	2	1	3 (3.8%)
Chemotherapy			
Neoadjuvant	19	36	55 (68.8%)
Post-operative	8	17	25 (31.3%)
Pathological therapeutic effect after NAC (*n* = 55)			
pCR	5	11	16
Non-pCR	14	25	39
Post-chemotherapy			
Endocrine	20	30	50 (62.5%)
CDK4/6 inhibitor	5	9	14 (17.5%)
Event			
Distant recurrence	0	5	5 (6.3%)
Death	0	3	3 (3.8%)

*^1^ For simplicity, the following classification was used. Premenopausal: 50 years old or younger; postmenopausal: older than 50 years; *^2^ *p* = 0.004, odds ratio: 4.71 (95% CI: 1.69–13.13), hazard ratio of 1.96 (95% CI: 1.30–2.96); ER: estrogen receptor; PgR: progesterone receptor; HER2: human epidermal growth factor receptor2; NAC: neoadjuvant chemotherapy; pCR: pathological complete response.

**Table 2 curroncol-32-00441-t002:** Comparison of NAC effects by subtype.

	Triple-Negative Type (*n* = 22)	Luminal Type (*n* = 33)
Clinical chemotherapy effect, no (%)		
cCR	11 (50.0)	5 (15.1)
cPR	11 (50.0)	27 (81.8)
cSD	0 (0)	1 (3.0)
cPD	0 (0)	0 (0)
Pathological chemotherapy effect, no (%) *		
pCR	13 (59.1) *	3 (9.1) *
non-pCR	9 (40.9)	30 (90.9)

* Odds ratio (OR): 14.4, *p*-value: 0.0001; CR: complete response; PR: partial response; SD: stable disease; PD: progressive disease; c: clinical; *p*: pathological.

**Table 3 curroncol-32-00441-t003:** Adverse events.

	Grade 1	Grade 2	Grade 3	Grade 4
Leukopenia (18.8%)	6 (7.5)	0 (0)	5 (6.3)	4 (5.0)
Neutropenia (18.8%)	5 (6.2)	1 (1.3)	5 (6.3)	4 (5.0)
Febrile neutropenia (8.8%)	0 (0)	0 (0)	7 (8.8)	0 (0)
Anemia (57.6%)	33 (41.3)	10 (12.5)	3 (3.8)	0 (0)
Thrombocytopenia (22.6%)	16 (20.0)	1 (1.3)	0 (0)	1 (1.3)
aspartate aminotransferase (58.8%)	40 (50.0)	6 (7.5)	1(1.3)	0 (0)
Anorexia (27.6%)	20 (25.0)	1 (1.3)	1 (1.3)	0 (0)
Nausea (37.6%)	24 (30.0)	5 (6.3)	1 (1.3)	0 (0)
Constipation (25.1%)	15 (18.8)	5 (6.3)	0 (0)	0 (0)
Diarrhea (3.8%)	3 (3.8)	0 (0)	0 (0)	0 (0)
Fatigue (43.8%)	29 (36.3)	4 (5.0)	2 (2.5)	0 (0)
Arthralgia (57.6%)	40 (50.0)	5 (6.3)	1 (1.3)	0 (0)
Myalgia (70.1%)	50 (62.5)	5 (6.3)	1 (1.3)	0 (0)
Peripheral neuropathy (55.0%)	34 (42.5)	10 (12.5)	0 (0)	0 (0)
Edema (16.3%)	10 (12.5)	3 (3.8)	0 (0)	0 (0)
Eczema (8.8%)	7 (8.8)	0 (0)	0 (0)	0 (0)
Stomatitis (10.1%)	7 (8.8)	1 (1.3)	0 (0)	0 (0)
Lung infection (7.5%)	0 (0)	2 (2.5)	4 (5.0)	0 (0)
Fever (12.5%)	8 (10.0)	2 (2.5)	0 (0)	0 (0)
Dysgeusia (17.5%)	12 (15.0)	2 (2.5)	0 (0)	0 (0)
Facial nerve disorder (1.3%)	0 (0)	1 (1.3)	0 (0)	0 (0)
Headache (3.8%)	3 (3.8)	0 (0)	0 (0)	0 (0)
Alopecia (100%)	0 (0)	80 (100.0)	0 (0)	0 (0)

## Data Availability

The data presented in this study are available on request from the corresponding author.

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
