# Peer review of "Efficacy and Safety of Dose-Dense Chemotherapy in Breast Cancer: Real Clinical Data and Literature Review"

_curroncol, 2025, doi:10.3390/curroncol32080441_

Round 1
Reviewer 1 Report
Comments and Suggestions for Authors
Efficacy and Safety of Dose-dense Chemotherapy in Breast 2
Cancer: Real Clinical Data and Literature Review
The research paper shows the efficacy and safety of dose-dense chemotherapy (ddCT) in breast cancer patients. Dose-dense chemotherapy, which demonstrates chemotherapy at shorter intervals without limiting the dose, shows improved patient outcomes. The manuscript looks promising for readers, clinicians and breast cancer researchers. The authors performed survival analysis and showed disease-free survival (DFS) and overall survival (OS) in patients treated with dose dense chemotherapy. However, the study showed toxicities and hematological side effects, which can be rectified by growth factor support. The literature review details also strengthen the findings and prove that ddCT has become a standard approach in high-risk early breast cancer patients. By incorporating these suggestions, the clarity and overall quality of the manuscript will be enhanced.
Minor Comments:
- Authors should cite and discuss these papers in the introduction section- DOI: 1200/JCO-24-01875, and DOI: 10.1016/j.lanepe.2024.101162. Also add 2-3 citations discussing the breast cancer recurrence and survival in the introduction section.
- The study lacks database information, and the number of studies reviewed. Please include.
- Authors should provide clear legends for the tables.
- A separate paragraph including detailed statistical analysis should be included.
- Authors mentioned Kaplan-Meier survival curve analysis in material and methods section but did not provide any figures. They should include the Kaplan-Meier survival figure, if possible.
- Authors should also add some limitations for dose dense chemotherapy based on elderly patients or organ dysfunction patients etc.
- How does the dose-dense chemotherapy compare with new therapies such as immunotherapy and CDK4/6 inhibitors? Please mention it in the discussion section.
- Please specify the difference between Grade 3 and 4 incidences.
Author Response
Thank you very much for taking the time to review this manuscript. We apologize for the numerous corrections, but we would appreciate your confirmation.
- Authors should cite and discuss these papers in the introduction section- DOI: 1200/JCO-24-01875, and DOI: 10.1016/j.lanepe.2024.101162. Also add 2-3 citations discussing the breast cancer recurrence and survival in the introduction section.
Thank you for pointing this out. I have added several papers to the introduction.
- The study lacks database information, and the number of studies reviewed. Please include.
- Authors should provide clear legends for the tables.
I improved as much as possible.
- A separate paragraph including detailed statistical analysis should be included.
- Authors mentioned Kaplan-Meier survival curve analysis in material and methods section but did not provide any figures. They should include the Kaplan-Meier survival figure, if possible.
We considered DFS and OS, but there were few events and no significant findings were obtained from the Kaplan-Meier curve. I have attached a graph showing DFS divided into premenopausal and postmenopausal groups.
- Authors should also add some limitations for dose dense chemotherapy based on elderly patients or organ dysfunction patients etc.
- How does the dose-dense chemotherapy compare with new therapies such as
immunotherapy and CDK4/6 inhibitors? Please mention it in the discussion section.
I added in the discussion.
- Please specify the difference between Grade 3 and 4 incidences.
Grade 4 adverse events were few, and only the frequency was separated from Grade 3 and recorded separately.

Reviewer 2 Report
Comments and Suggestions for Authors
Summary:
This manuscript presents a retrospective evaluation of dose-dense (dd) chemotherapy in breast cancer patients treated between 2020 and 2024, with a focus on its efficacy in triple-negative breast cancer (TNBC). The findings contribute meaningfully to the ongoing discussion regarding dd regimens and their outcomes in high-risk subtypes. However, the manuscript requires substantial revision before it can be considered for publication. Key issues include lack of clarity regarding the study cohort, insufficient statistical analysis, inconsistent use of abbreviations, and imbalanced manuscript structure.
Major Comments:
- Cohort Definition and Data Presentation:
- While the study initially refers to a total cohort of 80 patients, only 55 are included in the primary analysis (Table 2). This discrepancy should be addressed in the Methods section with clearly defined inclusion and exclusion criteria. If the remaining 25 patients were excluded due to not receiving neoadjuvant chemotherapy (NAC) or due to early discontinuation from severe adverse effects, this must be explicitly detailed in the Methods section. For better clarity and logical progression, consider placing Table 3 (appearing to include all 80 patients) within the Methods section to help illustrate the exclusion process, or at least position it before Table 2, and use markers such as asterisks or footnotes to denote excluded cases. Additionally, Table 1 should reflect only the analyzed population.
- The manuscript would benefit from including additional subgroup analyses (e.g., pre- vs. post-menopausal status), as these may reveal further insights into treatment effectiveness.
- Manuscript Structure and Content Balance:
- The Introduction is too brief and lacks sufficient context regarding the rationale and background for the use of dd chemotherapy. It should be expanded to better frame the study objectives.
- In contrast, the Discussion is overly long and descriptive. It would benefit from reorganization—grouping studies with similar findings and integrating them more analytically rather than descriptively.
- Some paragraphs summarize other studies extensively without linking them clearly to the current findings, while others state results with minimal contextualization. For example, the extended discussion on scalp cooling devices is not directly relevant, as these devices were not used in the study. This section should be significantly reduced or omitted.
- Critical Evaluation of Literature:
- When comparing the study's outcomes with prior trials (e.g., lines 243–247), the authors should offer hypotheses to explain differences, such as variations in treatment protocols, cancer stages, or patient characteristics. Currently, the discussion lacks this level of critical engagement.
- Language and Formatting Issues:
- Lines 89–108 largely reiterate the contents of the corresponding table and, mostly, do not provide additional interpretation or insights. This descriptive text is redundant and could be removed or significantly condensed to avoid unnecessary repetition.
- There are minor grammatical and formatting issues (e.g., “the” is misspelled or appears in a different font on lines 244 and 267).
- Lines 289 and 299 contain incomplete or unclear sentences that require revision.
- Line 353 begins a paragraph with “But,” which is stylistically inappropriate in formal writing and should be merged with the preceding paragraph.
- Tables are lacking the titles.
- Several abbreviations are either repeated unnecessarily (e.g., DFS, OS) or are not defined at all (e.g., CMF, CIA). Additionally, some terms are defined once but not consistently used throughout the text (e.g., NAC). A comprehensive and consistent approach to abbreviation usage is needed, including the abbreviation list.
- Supportive Care Considerations:
- The study notes that pegfilgrastim may cause more side effects in certain patients. The authors should discuss why subcutaneous injection of filgrastim was not used as an alternative and, if data are available, consider comparing outcomes between the two agents.
- Where was Kaplan-Meier survival curve used?
- The guidelines for defining pCR and non-pCR or partial response should be clearly described in the methods section.
Minor Comments:
- Ensure all defined abbreviations are actually used in the text, and avoid defining abbreviations that appear only once.
- Consider revising overly long sentences for clarity and readability.
Recommendation:
Major Revision
The study has merit and presents important findings on the use of dose-dense chemotherapy in breast cancer, particularly in TNBC. However, significant revisions are needed to improve the manuscript’s clarity, analytical depth, and presentation. I encourage the authors to address the points above in a thorough revision.
Author Response
Thank you very much for taking the time to review this manuscript. We have made significant revisions taking into consideration your comments. We apologize for the numerous corrections, but we would appreciate your confirmation.
Major Comments:
#1. Cohort Definition and Data Presentation:
While the study initially refers to a total cohort of 80 patients, only 55 are included in the primary analysis (Table 2). This discrepancy should be addressed in the Methods section with clearly defined inclusion and exclusion criteria. If the remaining 25 patients were excluded due to not receiving neoadjuvant chemotherapy (NAC) or due to early discontinuation from severe adverse effects, this must be explicitly detailed in the Methods section. For better clarity and logical progression, consider placing Table 3 (appearing to include all 80 patients) within the Methods section to help illustrate the exclusion process, or at least position it before Table 2, and use markers such as asterisks or footnotes to denote excluded cases. Additionally, Table 1 should reflect only the analyzed population.
The manuscript would benefit from including additional subgroup analyses (e.g., pre- vs. post-menopausal status), as these may reveal further insights into treatment effectiveness.
Thank you for pointing this out. I agree with this point.
Adverse events, DFS, and OS were investigated in all 80 patients who underwent dose-dense therapy.
For the 55 patients who underwent neoadjuvant chemotherapy, pathological complete response was recorded in Table 2.
Figure 1 has been added.
#2. Manuscript Structure and Content Balance:
The Introduction is too brief and lacks sufficient context regarding the rationale and background for the use of dd chemotherapy. It should be expanded to better frame the study objectives.
In contrast, the Discussion is overly long and descriptive. It would benefit from reorganization—grouping studies with similar findings and integrating them more analytically rather than descriptively.
Some paragraphs summarize other studies extensively without linking them clearly to the current findings, while others state results with minimal contextualization. For example, the extended discussion on scalp cooling devices is not directly relevant, as these devices were not used in the study. This section should be significantly reduced or omitted.
I added previous papers to the Introduction and explained the necessity of this study.
I also shortened the Discussion and deleted unnecessary information about devices.
#3. Critical Evaluation of Literature:
When comparing the study's outcomes with prior trials (e.g., lines 243–247), the authors should offer hypotheses to explain differences, such as variations in treatment protocols, cancer stages, or patient characteristics. Currently, the discussion lacks this level of critical engagement.
Although the number of cases is small, we have included as much information as possible in the Discussion section of this study.
#4. Language and Formatting Issues:
Lines 89–108 largely reiterate the contents of the corresponding table and, mostly, do not provide additional interpretation or insights. This descriptive text is redundant and could be removed or significantly condensed to avoid unnecessary repetition.
There are minor grammatical and formatting issues (e.g., “the” is misspelled or appears in a different font on lines 244 and 267).
Lines 289 and 299 contain incomplete or unclear sentences that require revision.
Line 353 begins a paragraph with “But,” which is stylistically inappropriate in formal writing and should be merged with the preceding paragraph.
Tables are lacking the titles.
Several abbreviations are either repeated unnecessarily (e.g., DFS, OS) or are not defined at all (e.g., CMF, CIA). Additionally, some terms are defined once but not consistently used throughout the text (e.g., NAC). A comprehensive and consistent approach to abbreviation usage is needed, including the abbreviation list.
Adverse events are listed by frequency and severity, and cases that required care, although rare, are also listed. (line162-)
Lines289 and 299(in the Initial manuscript) have been changed (line263- and line272).
I corrected text, conjunctions, and Table title.
Abbreviations have been organized.
#5. Supportive Care Considerations:
The study notes that pegfilgrastim may cause more side effects in certain patients. The authors should discuss why subcutaneous injection of filgrastim was not used as an alternative and, if data are available, consider comparing outcomes between the two agents.
In dose-dense chemotherapy, pegfilgrastim is added to all cases. In cases where pegfilgrastim caused thrombocytopenia, febrile neutropenia also occurred, so a short-acting G-CSF was added as an adjunct.
Initially, we considered the thrombocytopenia to be caused by the chemotherapy itself.After suspecting pegfilgrastim-induced thrombocytopenia, we switched to a 3-week regimen. However, prophylactic use of short-acting G-CSF was not feasible under the insurance system, so it was not administered.
Nevertheless, since bone marrow suppression no longer occurred with the 3-week regimen, chemotherapy was completed without the use of short-acting G-CSF.
#6. Where was Kaplan-Meier survival curve used?
We considered DFS and OS, but there were few events and no significant findings were obtained from the Kaplan-Meier curve. I have attached a graph showing DFS divided into premenopausal and postmenopausal groups.
#7. The guidelines for defining pCR and non-pCR or partial response should be clearly described in the methods section.
I have added the definition to the Method section.
#8. Minor Comments:
Ensure all defined abbreviations are actually used in the text, and avoid defining abbreviations that appear only once.
Consider revising overly long sentences for clarity and readability.
Those points have been corrected.

Round 2
Reviewer 2 Report
Comments and Suggestions for Authors
The authors have addressed all the questions raised during the review process. While I still believe that the methods section and the language could be improved, this is a minor issue. Overall, I am satisfied with the revisions made and approve the current version of the manuscript for publication, after language review.
Comments on the Quality of English LanguageThe quality of the English language should be improved.
As a small example, the sentence"We investigated differences in pCR rates after NAC between TNBC and Luminal type in premenopausal and postmenopausal patients. (Figure 1)", seems incomplete.
An improved version would be as follows: The differences in pCR rates after NAC between TNBC and luminal types in premenopausal and postmenopausal patients are shown in Figure 1.